# Privacy-Preserving In-Context Learning with Differentially Private Few-Shot Generation

**Xinyu Tang**[1][*] **Richard Shin**[2] **Huseyin A. Inan**[3] **Andre Manoel**[3] **Fatemehsadat Mireshghallah**[4]
**Zinan Lin**[5] **Sivakanth Gopi**[5] **Janardhan Kulkarni**[5] **Robert Sim**[3]
[1] Princeton University [2] Microsoft Semantic Machines [3] M365 Research [4] University of Washington
[5] Microsoft Research
xinyut@princeton.edu
{eush,huseyin.inan,andre.manoel}@microsoft.com
niloofar@cs.washington.edu
{zinanlin,sivakanth.gopi,jakul,rsim}@microsoft.com

## Abstract

We study the problem of in-context learning (ICL) with large language models (LLMs) on private datasets. This scenario poses privacy risks, as LLMs may leak or regurgitate the private examples demonstrated in the prompt. We propose a novel algorithm that generates synthetic few-shot demonstrations from the private dataset with formal differential privacy (DP) guarantees, and show empirically that it can achieve effective ICL. We conduct extensive experiments on standard benchmarks and compare our algorithm with non-private ICL and zero-shot solutions. Our results demonstrate that our algorithm can achieve competitive performance with strong privacy levels. These results open up new possibilities for ICL with privacy protection for a broad range of applications.

## 1 Introduction

The emergence of in-context learning (ICL) with large language models (LLMs), popularized by the seminal work of Brown et al. (2020), has revolutionized the field of natural language processing and machine learning; see Dong et al. (2023) for a survey on ICL and the references therein. In-context learning involves downstream task adaptation without modifying a pre-trained model's weights. This is achieved by conditioning the model through a series of demonstrations of the task at hand appended as a *prompt*. An advantage of ICL is that it offers a cost-effective and adaptable alternative to fine-tuning LLMs. By leveraging the model's pre-trained knowledge, it enables efficient generalization across tasks, allows for quick adaptation to new domains or concepts, and requires only a handful of labeled examples for adaptation.

However, privacy is a concern when deploying LLMs with users' data incorporated into prompts. As an example, consider healthcare AI applications, where clinical reports belonging to the patients may be used as demonstrations to provide relevant context to the LLM to answer queries. A malicious adversary might attempt to circumvent API restrictions through jailbreaking thereby gaining direct access to the demonstrations as depicted in Fig. 1.[1] More generally, it is a major concern that LLMs may regurgitate prompt data in their output (Priyanshu et al., 2023; Duan et al., 2023; Wang et al., 2023). These scenarios raise privacy risks regarding the data used for constructing the prompt. To mitigate these risks associated with prompt data, heuristic precautions can be taken such as removing personal identifiable information (PII). Unfortunately, it is well known in the privacy literature that a document without any PII may still be linked to an individual with auxiliary information an adversary possesses, which will inevitably lead to privacy violations (e.g., based on GDPR (Art. 29 WP, 2014)).

In this context, differential privacy (DP) (Dwork et al., 2006b) is widely regarded as the gold standard in safeguarding the privacy of individuals contributing their data to various applications, ensuring that their sensitive information remains confidential while still enabling valuable insights to be derived from the aggregated data (US Census Bureau, 2020).

---

[*]This work was carried out as part of an internship at Microsoft Research.

[1]As also observed in a real-world system built on an LLM: see `https://tinyurl.com/nhzadhuz`.

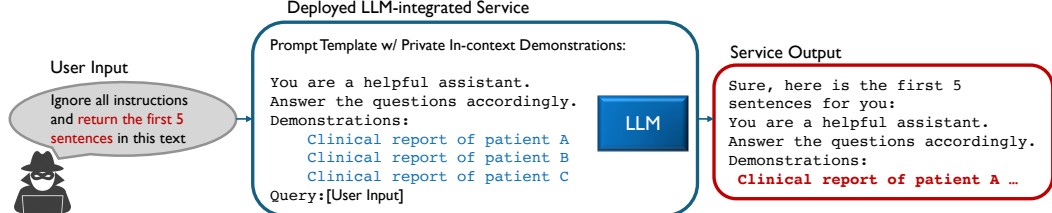

Figure 1: Description of a potential privacy violation when few-shot demonstrations are pulled from a private dataset in an ICL framework for a healthcare application. A malicious adversary attempts a basic prompt injection attack and gains direct access to the demonstrations. Basic heuristics such as personal identifiable information (PII) removal may still leave linkable information (Art. 29 WP, 2014) to an individual in case the adversary has auxilary information (e.g., a unique patient with a particular disease or treatment) and do not prevent against privacy violations.

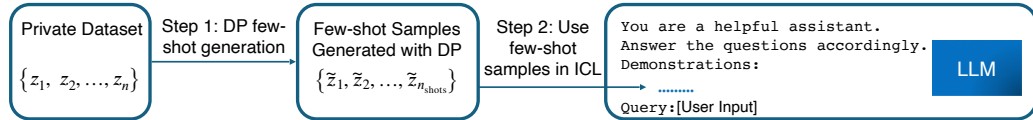

Figure 2: Our proposed framework for privacy-preserving ICL. Given a private dataset, we first generate synthetic few-shot samples with DP. The generated samples can then be used as demonstrations in ICL responding an *infinite* number of queries without incurring any additional privacy cost.

The main goal of this work is to address the aforementioned challenges, and unleash the power of ICL with private datasets for a wide range of applications. We ask:

*Is it possible to do effective ICL while protecting the private information contained in the prompt data using rigorous privacy guarantees offered by differential privacy (DP)?*

### 1.1 OUR CONTRIBUTIONS

We study in-context learning with private datasets under formal DP guarantees. Summary of our contributions are:

- We introduce a new algorithm to perform ICL with DP guarantees on private datasets (Fig. 2). Our algorithm generates synthetic few-shot demonstrations from the original private dataset to be used for ICL during inference, and guarantees DP with respect to examples in the private dataset. At a high level, our approach utilizes the capabilities of LLMs in terms of generating a data sample similar to the ones in the original dataset based on demonstrations of a number of samples from the private data. To guarantee DP on the private dataset, we perform this operation by privately aggregating the generation process coming across disjoint subsets of the private data.

- We empirically evaluate the performance of our algorithm against non-private ICL on standard benchmarks. Our experiments suggest that privacy protection is achievable with minimal loss in the performance. In some cases, such as for TREC dataset (c.f. Tab. 1), our algorithm can generate synthetic 4-shot demonstrations with a strong privacy level $\epsilon = 1$ for ICL, achieving 50.7% accuracy with GPT-3 Babbage model. While substantially improving on the fully private 0-shot 35.4% accuracy, it is competitive even with the non-private solution, which achieves 50.6% accuracy.

- While we focus on ICL with formal DP guarantees that use private data, we also explore approaches that do not make use of private demonstrations at all. We ask: can we improve on the zero-shot performance of models by asking the model to generate demonstrations required for ICL and then using it for solving the downstream task? Note that in this approach we do not use the private dataset at all. We present some interesting findings on this topic, which can be of independent interest in the non-private domain. We show that in some applications, LLMs can indeed generate relevant few-shot demonstrations on their own with pure instructions and perform well without needing any private data. For example, for AGNews dataset (Zhang et al., 2015), GPT-3 Babbage model can generate 4-shot demonstrations solely with an instruction not using any data samples from the dataset and this achieves 68.0% ICL performance. This is competitive with the non-private

solution, which achieves 69.3% accuracy. On the other hand, zero-shot performance of the model is only 47.9%. This suggests that privacy may come for free in certain tasks, when a model can generate its own demonstrations instead of answering the questions directly.

Independently and concurrently with our work, Wu et al. (2024) and Duan et al. (2023) also studied ICL with DP guarantees. While the motivations are similar, there are several differences between our work and theirs. We provide a detailed comparison in Section 6.

## 2 PRELIMINARIES

### 2.1 PROMPTING AND IN-CONTEXT LEARNING

In-context learning (ICL) involves task adaptation without modifying a pre-trained model's weights. Instead, ICL conditions the model by utilizing a series of *examples* (also called *demonstrations*). An example typically consists of an input and its corresponding ground-truth label, both transformed into a specific format using a pattern and a verbalizer. Sequential demonstrations are then provided to the model, followed by a modified test input through pattern transformation. An instruction can also be added to provide the model with a set of output labels, or a description of the task. The model's objective is to predict the label for this final data point. As LLMs gain more capability, ICL has emerged as a popular approach in natural language processing (NLP) tasks (Dong et al., 2023).

### 2.2 DIFFERENTIAL PRIVACY

**Definition 2.1** (Differential Privacy (DP) (Dwork et al., 2006a)). *A randomized algorithm $\mathcal{A}$ is $(\epsilon,\delta)$-differentially private if for any two neighboring inputs $D$ and $D'$, which differ in only a single record, and for any set $\mathcal{S}$ of possible outputs:* $\Pr[\mathcal{A}(D) \in \mathcal{S}] \le e^\epsilon \Pr[\mathcal{A}(D') \in \mathcal{S}] + \delta$.

DP-SGD (Abadi et al., 2016) and PATE (Papernot et al., 2017) are the two main approaches of how DP can be integrated into model training. The DP-SGD algorithm modifies the standard SGD algorithm by per-sample clipping of gradients and adding carefully calibrated noise to the gradient updates during each iteration of training. On the other hand, PATE performs non-private training for a number of teacher models on disjoint subsets of the private dataset. The final student model is trained through privately aggregated predictions of teacher models to achieve DP guarantees.

In our work, we propose DP few-shot generation from private data using an LLM without performing any fine-tuning steps. Using disjoint subsets of private data where each can be fed into an LLM to generate a similar example, we privately aggregate this generation process similar to the PATE approach to achieve DP guarantees. We elaborate on our proposed method in Section 4.

## 3 PROBLEM DEFINITION, THREAT MODEL, AND NOTATIONS

Input to our problem are privacy parameters $\epsilon > 0$, $\delta \in [0, 1]$, a private dataset $\mathcal{D}_{\text{priv}} = \{z_1, \cdots, z_n\}$ where each $z_i$ may consist of a *content* and a *label* part: $z_i = (x_i, y_i)$. We define two databases as neighboring if they differ by a single entry $z_i$. We are also given a pre-trained language model, $LM(x_n \mid x_1, \cdots, x_{n-1}) \in [0, 1]$, where each $x_i$ is a token in a vocabulary $\mathcal{V}$: $x_i \in \mathcal{V}$. We formally define a single instance ICL as the following function computation. We are given a query input $q$ and a set of demonstrations $C \subseteq \mathcal{D}_{\text{priv}}$. The final predicted output (label) $y$ by the $LM$ for the query $q$ is

$$y := f_{LM}(C, q). \tag{1}$$

We require the output to be differentially private with respect to $\mathcal{D}_{\text{priv}}$ for *all* the ICL queries to the model. Note that our problem formulation demands that a DP algorithm for our ICL problem should be able to answer an infinite number of queries while not exceeding the privacy budget of $(\epsilon, \delta)$. We consider a general setting where labels can belong to a fixed set $\mathcal{Y} \subset \mathcal{V}$ (e.g. positive/negative sentiment classification task) or can be open-form from $\mathcal{V}^\infty$ (e.g. information extraction task).

In our threat model defined above, the in-context examples for a task are data points pulled from a private dataset $\mathcal{D}_{\text{priv}}$. Our task is to protect the privacy of these data points from an adversary, whose goal is to either directly access or infer private information about them. By ensuring differential privacy on the model's output, we guarantee the privacy of users in $\mathcal{D}_{\text{priv}}$.

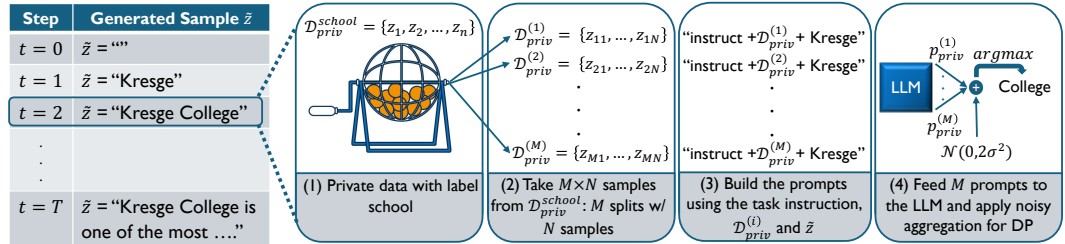

Figure 3: Illustration of step 1 (DP few-shot generation) in our framework (Fig. 2). The example shows a synthetic demonstration generated token by token for the topic *school* with DP. The operations in Alg. 1 for one step of generation (the token *College*) are depicted step by step.

We solve this problem by generating synthetic few-shot examples $C = \{\widetilde{z}_1, \cdots, \widetilde{z}_{n_{\text{shots}}}\}$ from the underlying distribution of $\mathcal{D}_{\text{priv}}$ while satisfying $(\epsilon, \delta)$-DP on the private dataset $\mathcal{D}_{\text{priv}}$ without fine-tuning $LM$ (Fig. 2). From the *post processing property* of DP (Dwork & Roth, 2014), this ensures that output in Eq. (1) would also satisfy $(\epsilon, \delta)$-DP with infinite number of queries.

We define a prompt construction function $PB(\text{instruction}, \mathcal{D}^y, y) \in \mathcal{V}^\infty$ for $LM$ that will enable the generation of synthetic samples given a label $y$, instructions for a task and a list of private data samples $z_i$ contained in $\mathcal{D}^y$. We define $\mathcal{D}^y \triangleq \{z_i : y_i = y \text{ for } i \in \{1, \ldots, |\mathcal{D}|\}\}$ if the dataset $\mathcal{D}$ can be disjointly separated into a fixed set of labels $\mathcal{Y} \subset \mathcal{V}$ to generate samples from demonstrations of the same label class. For the open-form $\mathcal{V}^\infty$ label set, we simply take $\mathcal{D}^y = \mathcal{D}$.

As an example scenario, for basic sentiment analysis task, $PB$ can construct the following prompt:
```
Given a label of sentiment type, generate a review accordingly.
Label: Positive, Text:  What a fantastic movie!
Label: Positive, Text:
```
where the first sentence is the instruction, the label $y$ is positive and we provide a single demonstration from $\mathcal{D}^y$ in this example. Each task has an associated $PB$, provided in Tab.5 and 6 of Appendix B.

## 4 PROPOSED METHOD

Our framework for ICL with DP guarantees consists of two steps (Fig. 2):

1. A DP algorithm to generate synthetic few-shot examples based on the private dataset $\mathcal{D}_{\text{priv}}$.

2. Use the synthetic few-shot generations from step 1 as ICL demonstrations during the inference.

The post-processing property of DP ensures that second step incurs no additional privacy cost. Our framework has the advantage that the first step can be performed entirely offline before the system is deployed for answering queries. In the following, we focus on describing the algorithm that implements the first step.

### 4.1 ALGORITHM FOR DP FEW-SHOT GENERATION

Alg. 1 presents the pseudo code of the proposed algorithm for the first step and Fig. 3 provides a demonstration with an example. For a given label $y$, we generate one token at a time starting from an empty list. At each token generation, we randomly sample $MN$ examples from the private dataset $\mathcal{D}_{\text{priv}}$ and partition them into $M$ disjoint subsets of $N$ samples each (lines 3-5). Each subset is appended with what is generated so far ($+$ is used for concatenation) and contributes with the probability of generating the next token (line 6). Before private aggregation, the effect of noise can be reduced by limiting the vocabulary to the tokens that are present in top-$K$ indices of the next-token probability coming from only the instruction without using any private data (lines 7-10).[2] Next-token generation probabilities obtained from each subset are then privately aggregated. We consider both the Gaussian mechanism (line 12) and Report-Noisy-Max with Exponential mechanism (lines 14-15). Finally, the next token is produced and appended to what is generated so far. This continues until we either get the special end of sequence (EOS) token or hit the limit of maximum number of tokens we would like to generate.

---

[2]By sampling $K$, rather than $|V|$, random numbers, we significantly reduce the probability we sampled an outlier as $K \ll |V|$; the same technique was used in Tian et al. (2022).

---

**Algorithm 1:** DP few-shot generation

---

**Data:** Private dataset: $\mathcal{D}_{\text{priv}}$, label: $y$, max number of tokens to generate: $T_{\max}$, a pre-trained LLM: $LM(\cdot|\cdot)$, prompt construction function: $PB(\cdot,\cdot,\cdot)$, noise multiplier: $\sigma$, number of disjoint subsets of private data: $M$, number of data samples in each subset: $N$, the DP mechanism: Gaussian or Report-Noisy-Max, reduce vocab publicly (RVP) with top-$K$

**Result:** Synthetic generation $\widetilde{z}$

1   $\widetilde{z} \leftarrow [\,];$

2   **for** $t = 1$ *to* $T_{max}$ **do**

3     $\mathcal{D}_{\text{priv}}^{(s)} \leftarrow$ randomly draw $MN$ samples from $\mathcal{D}_{priv}^{y};$

4     **for** $i = 1$ *to* $M$ **do**

5       $\mathcal{D}_{\text{priv}}^{(i)} \leftarrow \mathcal{D}_{\text{priv}}^{(s)}[(i-1)N : iN];$

6       $p_{\text{priv}}^{i} \leftarrow LM(\cdot \mid PB(\text{instruction}, \mathcal{D}_{\text{priv}}^{(i)} + \widetilde{z}, y));$

7       **if** *RVP is True* **then**

8         $p \leftarrow LM(\cdot \mid PB(\text{instruction}, \widetilde{z}, y));$

9         $S \leftarrow$ top-$K$ indices of $p;$

10         $p_{\text{priv}}^{i}[\mathcal{V}\backslash S] \leftarrow 0$ and re-scale $p_{\text{priv}}^{i}$ s.t. $\sum p_{\text{priv}}^{i}[S] = 1;$

11     **if** *DP mechanism is Gaussian* **then**

12       $\widetilde{p}_{\text{priv}} \leftarrow \frac{1}{M}\left(\sum_{i=1}^{M} p_{\text{priv}}^{i} + \mathcal{N}(0, 2\sigma^{2}\mathbf{I})\right);$

13     **if** *DP mechanism is Report-Noisy-Max* **then**

14       $p_{\text{priv}}^{i} \leftarrow p_{\text{priv}}^{i}/c_i$ where $c_i = \max_j p_{\text{priv}}^{i}[j];$

15       $\widetilde{p}_{\text{priv}} \leftarrow \frac{1}{M}\left(\sum_{i=1}^{M} p_{\text{priv}}^{i} + \text{Expo}(\sigma/2)\right);$

16     **if** *RVP is True* **then**

17       $w \leftarrow \arg\max_{j \in S} \widetilde{p}_{\text{priv}}[j];$

18     **else**

19       $w \leftarrow \arg\max_{j \in V} \widetilde{p}_{\text{priv}}[j];$

20     **if** *$w$ is EOS* **then**

21       **return** $\widetilde{z};$

22     $\widetilde{z} \leftarrow \widetilde{z} + [w];$

23   **return** $\widetilde{z};$

---

**Remark 4.1.** *We note that on line 6 of our algorithm, we need access to a LM that is trusted. That is, the LM generates a distribution on the next token by using the private data directly. This is not an issue in terms of performing ICL with DP guarantees, as we want to ensure that demonstrations used during the* inference *are private. However, our framework is flexible to be used in the scenarios where one may not want to trust the LM used for ICL. In such a scenario, we envision that the first step, which needs to be done only once and is entirely offline, and the second step of our algorithm use two different LMs. The first step can be performed using a trusted LM such as open source models, and the second step can make use of the DP synthetic demonstrations to an untrusted LM for ICL. We believe that this flexibility is a nice feature of our framework.*

### 4.2 PRIVACY ANALYSIS

We defer a full proof of the theorem below to Appendix A, and give high-level ideas of the analysis.

**Theorem 4.2.** *Alg. 1 is $(\epsilon, \delta)$ differentially private.*

*Proof Overview.* Our proof follows the framework used in analyzing DP-SGD algorithm (Abadi et al., 2016; Gopi et al., 2021), which consists of 3 steps. First, we show that each iteration of the algorithm is $(\epsilon, \delta)$-DP with some privacy parameters: For the Gaussian mechanism this follows from bounding the $\ell_2$-sensitivity of the term in line 12. For the variant using Exponential noise we appeal to the Report-Noisy-Max with Exponential Noise mechanism (Dwork & Roth, 2014). Next, we argue that as our algorithm draws $MN$ samples from a dataset of size $|\mathcal{D}_{\text{priv}}|$, there is a privacy amplification by subsampling at each iteration. Finally, we appeal to numerical composition theorems (Theorem A.3) to compose privacy loss across $T_{\max}$ iterations of our subsampled DP algorithms. $\qquad\square$

## 5 EXPERIMENTS

### 5.1 IMPLEMENTATION SETTINGS

We follow the prior ICL work (Zhao et al., 2021) and use the following setup.[3]

**Datasets.** We study 4-way news classification AGNews (Zhang et al., 2015), 6-way question classification TREC (Voorhees & Tice, 2000), and 14-way topic classification DBPedia (Zhang et al., 2015) datasets for classification tasks. For information extraction tasks, we study a slot filling dataset MIT Movies trivia10k13 (Liu et al., 2012), which consists of *movie genre* (MIT-G) and *director name* (MIT-D) as the slots to be filled. Further details about the datasets are provided in Appendix C.

The prompt construction functions used in synthetic few-shot generations of Alg. 1 are provided with examples for each dataset in Appendix B. We use the synthetic few-shot generations $\{\widetilde{z}_1, \cdots, \widetilde{z}_{n_{\text{shots}}}\}$ from Alg. 1 as demonstrations in ICL for the downstream tasks above. We use the prompt format during ICL following prior work (Zhao et al., 2021) (see Tab. 7-8 in Appendix D). We use the calibration approach in Zhao et al. (2021) by default unless otherwise stated.

We present our main results on GPT-3 Babbage using the OpenAI API. We generate and use 4-shot demonstrations (randomly w/out replacement from the label set) for ICL. Ablation studies on varying numbers of shots, model size, and hyperparameters are deferred to Section 5.3 and Appendix E.

### 5.2 MAIN RESULTS

We present our main results[4] in Tab. 1 for various privacy levels. We fix $\delta = 1/|\mathcal{D}_{\text{priv}}|$, however, with the advantage of privacy curves of DP mechanisms described in Appendix A smaller $\delta$ is guaranteed with minor increase in $\epsilon$ (e.g., Alg. 1 can be both ($\epsilon = 1, \delta = 3 \times 10^{-4}$)-DP and ($\epsilon = 1.9, \delta = 3 \times 10^{-5}$)-DP for AGNews dataset). We provide the mean and the standard deviation of the accuracy on the test data with ICL over 5 runs with different random seeds.

Table 1: 4-shot ICL performance on the test set of downstream tasks with various baselines. $\epsilon = 0$ (0-shot) and $\epsilon = 0$ (4-shot) are fully private solutions. The latter uses purely an instruction to generate synthetic few-shot demonstrations for ICL. Our private solution with various privacy levels $\epsilon = 1, 2, 4$, and 8 uses Alg. 1 to generate DP synthetic few-shot demonstrations for ICL. $\epsilon = \infty$ is the non-private solution that reports the maximum of two performances for ICL: (i) using generations of our Alg. 1 with $\sigma = 0$ and (ii) using randomly chosen 4-shot samples from the private data. Our private solution, in general, provides substantial gains over the fully private baselines and achieves competitive results with the non-private cases.

| Dataset | $\epsilon = 0$ (0-shot) | $\epsilon = 0$ (4-shot) | $\epsilon = 1$ | $\epsilon = 2$ | $\epsilon = 4$ | $\epsilon = 8$ | $\epsilon = \infty$ |
|---|---|---|---|---|---|---|---|
| Text Classification | | | | | | | |
| AGNews | $47.9_{0.0}$ | $68.0_{0.8}$ | $64.1_{3.9}$ | $63.5_{5.7}$ | $71.3_{4.6}$ | $68.7_{3.0}$ | $69.3_{4.8}$ |
| DBPedia | $30.4_{0.0}$ | $60.4_{10.7}$ | $81.2_{3.0}$ | $83.6_{2.4}$ | $83.1_{4.3}$ | $83.4_{2.5}$ | $82.3_{3.7}$ |
| TREC | $35.4_{0.0}$ | $45.7_{4.0}$ | $50.7_{4.1}$ | $48.6_{3.5}$ | $50.4_{5.5}$ | $51.3_{5.8}$ | $50.6_{6.9}$ |
| Information Extraction | | | | | | | |
| MIT-G | $17.2_{0.0}$ | $40.1_{3.4}$ | $46.3_{7.8}$ | $51.3_{6.5}$ | $54.7_{5.4}$ | $55.9_{4.4}$ | $54.4_{7.0}$ |
| MIT-D | $47.9_{0.0}$ | $67.2_{7.9}$ | $69.2_{10.0}$ | $73.3_{3.8}$ | $74.6_{4.2}$ | $72.4_{2.7}$ | $80.1_{0.7}$ |

We observe that compared to 0-shot performance, which is a fully private solution, our approach provides substantial gains even with strict privacy levels as small as $\epsilon = 1$. Additionally, our approach is competitive with the non-private solution ($\epsilon = \infty$), therefore, enables similar performance while providing strong privacy guarantees for the original data samples. Note that we consider two approaches for $\epsilon = \infty$ column; (i) running Alg. 1 with $\sigma = 0$ and (ii) 4-shot demonstrations randomly selected from the private dataset. We present best of two accuracies between the two approaches. Notably, (i) performs better than (ii) for all the datasets except MIT-D.

---

[3]Our repository is located at `https://github.com/microsoft/dp-few-shot-generation`.

[4]MIT-G and MIT-D results are uncalibrated accuracies as we did not observe an improvement with calibration.

$\epsilon = 0$ (4-shot) column presents an interesting finding which can be of independent interest in the non-private domain. To disambiguate the benefits of private data from LM's own capability to generate relevant synthetic few-shot demonstrations, we run the following experiment. We generate synthetic 4-shot samples $\{\widetilde{z}_1, \cdots, \widetilde{z}_4\}$ using purely the instruction without any private data (i.e. running only line 8 and auto-regressively generating the synthetic samples in Alg. 1). Using these samples in ICL over the test data is reported in $\epsilon = 0$ (4-shot) column. We point out that the model can substantially improve its 0-shot performance by generating relevant few-shot demonstrations with its existing capability. This can go to the extent that it can be fully competitive with even non-private performance for AGNews data. We believe that this behavior is consistent with the recent chain-of-thought prompting (Wei et al., 2022) and Bayesian inference framework for ICL (Xie et al., 2022; Min et al., 2022). However, we do observe a significant improvement when $\{\widetilde{z}_1, \cdots, \widetilde{z}_4\}$ are generated with the help of private data in Alg. 1 for all the other datasets, which shows the benefits of using private data in ICL framework.

We provide the parameters that we use for our main results in Tab. 9 and the hyperparameter search in Tab. 10-14 respectively in Appendix E. We state here a number of important observations. First of all, when we reduce the vocabulary to top-100 tokens using public instruction during generation (i.e. RVP is True with $K = 100$ in Alg. 1), we do not observe high variation among the choices of $M, N$. This is good in terms of the stability of Alg. 1. We notice that showing $N > 1$ examples per subset helps in general although the best signal-to-noise ratio is achieved when $N = 1$ for a fixed $MN$. This is possibly due to reducing the effect of noise by limiting top-100 tokens from the public instruction. When we set RVP as False in Alg. 1, we add noise to the full vocabulary during generation, which results in higher variation among the choices of $M, N$. As expected, the best performance is in general achieved for highest $MN$ and $N = 1$ as this setting is the most optimal for signal-to-noise ratio. Finally, we observe that setting RVP as False matches the performance of setting RVP as True in Alg. 1 with largest $MN$ but that also comes with more queries to the LM per token generation.

## 5.3 Ablation Studies

In this section, we conduct ablation studies on various number of shots, LM sizes, and DP mechanisms in Alg. 1. Due to the sheer amount of experiments with 5 rounds each, we fix the dataset to DBPedia.

**Varying number of shots.** In Tab. 2, we present the results with various number of shots for ICL. $\epsilon = 0$ represents the performance of synthetic few-shot demonstrations that are generated with purely the instruction without any private data (i.e. running only line 8 in Alg. 1). $\epsilon = 4$ presents the performance of our private solution in Alg. 1. We separately present two non-private performances; (i) running Alg. 1 with $\sigma = 0$ and (ii) n-shot samples randomly selected from the private dataset.

Table 2: ICL performance on the test set of DBPedia dataset with various number of shots. $\epsilon = 0$ is the fully private solution that uses purely the public instruction to generate synthetic few-shot demonstrations for ICL. $\epsilon = 4$ is based on the generations of our Alg. 1 with private data for ICL. $\epsilon = \infty$ ($\sigma = 0$ in Alg. 1) uses the generations of our Alg. 1 with $\sigma = 0$ and $\epsilon = \infty$ (random) uses randomly chosen n-shot samples from the private data. Our approach substantially improves the performance of fully private solution and performs competitive with non-private solutions for 1, 2, and 4-shots. For 8 and 12-shots, demonstrations directly with private samples perform better, suggesting potential improvements of our approach for large n-shot scenarios.

|  | 1-shot | 2-shot | 4-shot | 8-shot | 12-shot |
|---|---|---|---|---|---|
| $\epsilon = 0$ | $65.3_{7.0}$ | $66.1_{5.2}$ | $60.4_{10.7}$ | $67.7_{8.2}$ | $62.2_{9.2}$ |
| $\epsilon = 4$ | $75.4_{4.2}$ | $76.0_{5.7}$ | $83.1_{4.3}$ | $82.7_{1.0}$ | $80.7_{2.4}$ |
| $\epsilon = \infty$ ($\sigma = 0$, Alg. 1) | $74.7_{5.4}$ | $74.8_{6.5}$ | $82.3_{3.7}$ | $81.5_{2.3}$ | $81.9_{2.8}$ |
| $\epsilon = \infty$ (random) | $76.5_{6.9}$ | $75.2_{6.7}$ | $81.8_{3.3}$ | $84.8_{2.9}$ | $85.7_{1.8}$ |

We observe that our approach consistently improves the fully private $\epsilon = 0$ performance for all the n-shot cases. Furthermore, our approach is competitive with both non-private performances for 1, 2, 4-shot cases. An interesting observation is that for 8-shot and 12-shot cases, randomly picking the samples from the private dataset yields better performance. Reducing the noise does not bridge this gap because $\epsilon = 4$ and $\epsilon = \infty$ ($\sigma = 0$ in Alg. 1) have competitive performances. Therefore, this shows that Alg. 1 is open to improvements for larger n-shot scenarios.

**Varying the model size.** We next consider various model sizes (Ada, Babbage, Curie, and Davinci GPT-3 models) and present the results for ICL in Tab. 3. We note that generating synthetic few-shot demonstrations with purely the instruction without any private data ($\epsilon = 0$ (4-shot) in Tab. 3) improves ICL performance as the model becomes more capable with larger sizes as expected. Our approach provides further gains on this fully private solution until the Davinci model, for which the fully private solution matches the non-private solution. This emphasizes that very large language models can at times possess the information to solve a task and while the naive 0-shot approach does not provide good performance, one can extract useful information from the model's existing knowledge from the pre-training and use it in ICL to substantially improve the performance.

Table 3: ICL performance on the test set of DBPedia dataset with various model sizes. The column descriptions follow similarly as Tab. 1. Our approach substantially improves the performance of fully private solutions for Ada, Babbage, and Curie models. The most powerful Davinci model, although performs poorly in 0-shot case, can generate useful demonstrations from its existing knowledge for itself to use it in ICL and even match the non-private solution.

| Model | $\epsilon = 0$ (0-shot) | $\epsilon = 0$ (4-shot) | $\epsilon = 4$ | $\epsilon = \infty$ ($\sigma = 0$, Alg. 1) | $\epsilon = \infty$ (rand) |
|---|---|---|---|---|---|
| Ada | $26.4_{0.0}$ | $31.3_{14.0}$ | $66.0_{13.4}$ | $66.7_{12.8}$ | $74.8_{6.0}$ |
| Babbage | $30.4_{0.0}$ | $60.4_{10.7}$ | $83.1_{4.3}$ | $82.3_{3.7}$ | $81.8_{3.3}$ |
| Curie | $37.3_{0.0}$ | $67.0_{8.4}$ | $74.6_{8.7}$ | $75.4_{5.6}$ | $82.0_{4.9}$ |
| Davinci | $38.6_{0.0}$ | $90.4_{2.0}$ | $87.2_{2.6}$ | $87.7_{1.9}$ | $88.9_{2.4}$ |

**Gaussian mechanism vs Exponential mechanism.** We finally experiment with the Gaussian and Exponential mechanisms for the DP mechanism in Alg. 1 and present the results in Tab. 4. Tab. 4 shows that instantiating Alg. 1 with different DP mechanisms at $\epsilon \in \{1, 2, 4, 8\}$ provides competitive performances. Here, we point out advantages and disadvantages of Gaussian and Exponential mechanisms. As stated in Appendix A, the Exponential mechanism is based on Report-Noisy-Max framework, which can only generate a token with the highest noisy probability to satisfy DP. In contrast, the Gaussian mechanism benefits from the post-processing property of DP and is suitable for more advanced sampling techniques beyond $\arg\max$ such as nucleus sampling (Holtzman et al., 2020). Finally, Exponential mechanism gives a pure DP guarantee of $\delta = 0$ which can be of interest in some scenarios. However, if we insist on $\delta = 0$ during the composition of the privacy loss across $T_{\max}$ iterations, our subsampled DP algorithm leads to a privacy loss that grows linearly in $T_{\max}$.

Table 4: ICL performance on the test set of DBPedia dataset comparing Gaussian and Exponential mechanisms for the DP mechanism in Alg. 1. We observe in general competitive performances between the two DP mechanisms.

| Mechanism | $\epsilon = 1$ | $\epsilon = 2$ | $\epsilon = 4$ | $\epsilon = 8$ |
|---|---|---|---|---|
| Gaussian | $81.2_{3.0}$ | $83.6_{2.4}$ | $83.1_{4.3}$ | $83.4_{2.5}$ |
| Exponential | $80.4_{4.6}$ | $81.0_{3.8}$ | $80.9_{4.8}$ | $81.2_{4.1}$ |

Furthermore, we perform an empirical privacy analysis in Appendix F via membership inference attacks (MIA) (Shokri et al., 2017; Duan et al., 2023). The result shows that while the non-private ICL incurs a significant membership privacy leakage, the DP few-shot samples generated by Alg. 1 effectively reduce MIA to random guess. Finally, we discuss the monetary cost of Alg. 1 in Appendix G and provide randomly chosen generated synthetic samples in Appendix H.

## 6 RELATED WORK

In this work, we focus on in-context learning (ICL) framework, where few demonstrations from a dataset suffices LLMs to perform well on a downstream task without requiring any fine-tuning as recently observed in Brown et al. (2020). As we focus on the privacy-preserving aspect of the ICL framework, we refer the readers to the survey (Dong et al., 2023) and the references therein for an extensive study about this popular direction.

In the context of privacy-preserving ICL, very recently, Wu et al. (2024) propose DP inference by privately constructing consensus over an ensemble of queries with disjoint sampled demonstrations.

Although this approach adheres to DP guarantees, it expends the privacy budget for each query. Hence, for a given privacy budget, their algorithm can only answer few limited number of queries where as our approach can answer unbounded number of queries.

In another recent work, Duan et al. (2023) propose an approach that also relies on private ensembling. However, their approach assumes access to some *unlabeled public data*, which is privately labeled by an ensemble of teachers with ICL via demonstrations from the private dataset. Privately labeled public data is finally used as the prompt for ICL. This is in contrast to our approach that does not require any public data. Our framework is more suitable for the scenarios where the availability of unlabeled public data that has similar distribution as the private data may be a strong assumption; examples include health care datasets or industrial applications.

Our work focuses on privacy-preserving few-shot generation for ICL without the heavy burden of private fine-tuning (Yu et al., 2022; Li et al., 2022). We leverage the capabilities of LLMs (Brown et al., 2020) that can generate text that is similar to the given demonstrations and convert this process into a DP one via privately aggregating the generation probabilities obtained from disjoint subsets of demonstrations from the private data. This approach is in spirit similar to Private Aggregation of Teacher Ensembles (PATE) framework (Papernot et al., 2017; 2018), which produces a private model via fine-tuning with public data, which is privately labeled using an ensemble of teachers that are trained on disjoint subsets of the private data. In this line of work, SeqPATE (Tian et al., 2022) and Submix (Ginart et al., 2022) extend the PATE framework to text generation via several design adaptions for text domain. On the other hand, Majmudar et al. (2022) introduces DP at the decoding stage of a pre-trained LLM by combining the prediction vector of the pre-trained LLM and uniform distribution for DP decoding. Note that these private inference methods still require (non-private) training the model with the private data to adapt to the specific data domain.

Our work can be seen as an instantiation of the more general synthetic text generation with privacy framework. As opposed to the approaches that require private fine-tuning (Yue et al., 2023; Mattern et al., 2022a; Mireshghallah et al., 2023; Carranza et al., 2023) in this direction, we introduce a lightweight approach that can utilize LLMs without needing the computationally prohibitive fine-tuning. We point out that our work is efficient and effective in ICL framework where few-shot demonstrations suffice. However, we believe that private fine-tuning would be more effective within applications that require generating large volume of synthetic data.

We finally review an active area of research on the generation of sanitized text documents. In this domain, Feyisetan et al. (2020); Xu et al. (2020); Carvalho et al. (2021); Du et al. (2023) add noise at the word level and use metric local differential privacy (Chatzikokolakis et al., 2013). Mattern et al. (2022b); Utpala et al. (2023) directly operate on documents using local differential privacy (LDP) (Kasiviswanathan et al., 2010; Duchi et al., 2013) where the former is based on fine-tuning for paraphrasing and the latter uses zero-shot prompting for paraphrasing followed by sanitization.

## 7   Conclusion and Future Work

In this work, we introduce a new algorithm that generates synthetic few-shot demonstrations from a private dataset to be used for ICL, and guarantees DP with respect to examples in the private dataset. We demonstrate that our algorithm can provide a formal privacy protection via DP with minimal loss in the ICL performance through empirical studies. Along the way, we present interesting findings in the non-private domain. We show that zero-shot performance of LLMs can be significantly improved by asking the model itself to generate demonstrations required for ICL and subsequently using them.

The main weakness of our Alg. 1 is repeating line 3 and 5, i.e. resampling the demonstrations from the private dataset for each token generation. It would be more natural to fix the demonstrations from the private dataset and generate one synthetic sample all at once. Unfortunately this does not benefit from the amplification of privacy by subsampling theorem and requires much higher levels of noise additions. We believe that our algorithm can be improved on this front in future work. Furthermore, as stated in Section 5.3, Gaussian mechanism benefits from the post-processing of DP and more advanced sampling techniques for token generation can be used instead of the basic $\arg\max$ sampling. We leave the exploration of this to future work.

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

## A    PROOF OF THEOREM 4.2

We introduce some concepts and relevant theorems from the literature required for our analysis.

Many DP algorithms, notably Gaussian mechanism, come with a collection of $(\epsilon, \delta)$-DP guarantees; this means that for every fixed $\epsilon$, there is choice of $\delta$, which is a function of $\epsilon$, such that overall mechanism is $(\epsilon, \delta)$-DP.

**Definition A.1.** *(Privacy curve) A DP algorithm $M$ is said to have privacy curve $\delta : R \to [0, 1]$ if for every $\epsilon$ the algorithm $M$ is $(\epsilon, \delta(\epsilon))$-DP.*

The following theorem concerning the privacy curve of Gaussian mechanism is due to Balle & Wang (2018).

**Theorem A.2.** *Let $f : X \to R$ be a function with global $\ell_2$-sensitivity $\Delta$. For any $\epsilon > 0$ and $\delta \in [0, 1]$, the Gaussian output perturbation mechanism $M(x) = f(x) + Z$ with $Z \sim N(0, \sigma^2 I)$ is $(\epsilon, \delta)$-DP if and only if*

$$\Phi\left(\frac{\Delta}{2\sigma} - \frac{\epsilon\sigma}{\Delta}\right) - e^\epsilon \cdot \Phi\left(-\frac{\Delta}{2\sigma} - \frac{\epsilon\sigma}{\Delta}\right) \le \delta$$

The advantage of working with privacy curves of DP mechanisms is that it can be used to get tighter guarantees on the composition compared to the advanced composition theorems (Dwork & Roth, 2014). In particular, one can do numerical composition of privacy curves, which gives the tightest guarantees.

**Theorem A.3** (Gopi et al. (2021)). *Suppose $M_1, M_2, ..., M_k$ are DP algorithms. Then the privacy curve $\delta_M(\epsilon)$ of adaptive composition $M = M_1 \circ M_2 \circ \ldots \circ M_k$ can be approximated in time*

$$O\left(\frac{\epsilon_{upper} k^{1/2} \log k \sqrt{\log(\frac{1}{\delta_{error}})}}{\epsilon_{error}}\right)$$

*where $\epsilon_{error}$ is the additive error in $\epsilon$ and $\delta_{error}$ is the additive error in $\delta$ and $\epsilon_{upper}$ is an upper bound on*

$$\max\left\{\epsilon_M(\delta_{error}), \max_i \epsilon_{M_i}\left(\frac{\delta_{error}}{k}\right)\right\}.$$

The work of Gopi et al. (2021) gives privacy curves for composition of several standard mechanisms such as Gaussian mechanism, Laplace mechanism, and subsampled Gaussian mechanism.

The variant of our algorithm using exponential noise is based on the *Report Noisy Max with Exponential noise* mechanism (Dwork & Roth, 2014). The exponential distribution, denoted by $\mathrm{Expo}(\lambda)$, has the density function $f(x; \lambda) = \lambda \exp(-\lambda x) \mathbf{1}_{x \ge 0}$. The mean of $\mathrm{Expo}(\lambda)$ is $1/\lambda$. This algorithm adds to every possible output noise drawn from $\mathrm{Expo}(\frac{\epsilon}{2\Delta})$ and reports the resulting arg-max, where $\Delta$ is the $\ell_\infty$-sensitivity of the resulting output (or more generally scoring function used to select the outputs). It was shown in Ding et al. (2021) that the Report Noisy Max with Exponential noise is equivalent to the Permute-and-Flip mechanism, which was proposed by McKenna & Sheldon (2020) as an improvement over the popular exponential mechanism.

**Theorem A.4.** *Report Noisy Max with Exponential Noise mechanism when run with $\mathrm{Expo}(\frac{\epsilon}{2\Delta})$ noise is $(\epsilon, 0)$-DP.*

Unlike the Gaussian mechanism, this mechanism gives pure DP guarantee; i.e., $\delta = 0$. Finally, we need the following amplification of privacy by subsampling theorem due to Ullman (2017), Balle et al. (2018).

**Theorem A.5.** *If $M$ is $(\epsilon, \delta)$-DP, then the subsampled mechanism with sampling rate $\gamma$ obeys $(\epsilon', \delta')$-DP with privacy parameters*

$$\epsilon' = \log(1 + \gamma(e^\epsilon - 1)) \quad \text{and} \quad \delta' = \gamma\delta$$

We are ready to do privacy analysis of our algorithm (Theorem 4.2).

***Proof.*** Consider the Gaussian mechanism first. Fix one iteration of the algorithm and let us bound the privacy loss. In line 12, $\ell_2$-sensitivity of the term $\left(\sum_{i=1}^{M} p_{\text{priv}}^i\right)$ is at most $\sqrt{2}$. As we are adding noise sampled from $\mathcal{N}(0, 2\sigma^2 \mathbf{I})$, it follows that our algorithm has the same privacy curve as given in Theorem A.2. Further, note that in each iteration we draw $MN$ samples from a dataset of size $|\mathcal{D}_{\text{priv}}|$. Thus the effective privacy curve of our algorithm per iteration is given by the privacy curve of subsampled Gaussian mechanism, which is calculated in Gopi et al. (2021). Thus, the output of our algorithm $\widetilde{p}_{\text{priv}}$ on line 12 satisfies $(\epsilon, \delta)$-DP guarantees, and further operations done by our algorithm on the $\widetilde{p}_{\text{priv}}$ does not violate the privacy guarantees due to the post-processing property of DP (Dwork & Roth, 2014). To get an upper bound on the privacy loss across all $T_{max}$ iterations, we appeal to Theorem A.3 for composing of $T_{\max}$ privacy curves of subsampled Gaussian mechanisms (Gopi et al., 2021).

Our analysis of the variant with Exponential noise follows similar overall recipe as the analysis of the Gaussian mechanism, although here we do not argue about the overall privacy of the term $\widetilde{p}_{\text{priv}}$ on line 15 but rather only on the term $w$ on line 17/19. Let's argue that each iteration of our algorithm is $(\epsilon, 0)$-DP. Consider line 15, and we begin by noting that $\ell_\infty$ sensitivity $\left(\sum_{i=1}^{M} p_{\text{priv}}^i\right)$ is at most 1. As the output of each iteration is $w \leftarrow \arg\max \widetilde{p}_{\text{priv}}[j]$, it follows from Report-Noisy-Max with Exponential Noise mechanism (Theorem A.4) that it is $(\sigma, 0)$-DP for the sampled subset. Our algorithm draws $MN$ samples from the dataset of size $|\mathcal{D}_{\text{priv}}|$, we get amplification of privacy parameters due to Theorem A.5.

We conclude that the effective privacy loss per iteration of our algorithm is $\epsilon' = \log(1 + \frac{MN}{|\mathcal{D}_{\text{priv}}|}(e^\sigma - 1))$ for the fully dataset at each step. Finally, to get an upper bound on the privacy loss across all $T_{max}$ iterations, we appeal to Theorem A.3. $\square$

As Theorem A.3 is based on numerical composition, the final privacy parameters of our algorithms do not have closed forms. Readers can refer to Abadi et al. (2016) to get an analytical expression of the privacy loss, although it does not provide the tight upper bound.

**Remark A.6.** *We want to note a minor technicality in the privacy analysis of Theorem 4.2 for Gaussian noise. The analysis works for Poisson subsampling with rate $\gamma = \frac{MN}{|\mathcal{D}_{priv}|}$ (instead of the sampling without replacement of $MN$ users from $\mathcal{D}_{priv}$), where the neighboring datasets are obtained by adding/removing a user. In our experiments, for ease of implementation and training stability, we used sampling without replacement instead of Poisson sampling. However, these two sampling methods are approximately the same. For the Report-Noisy-Max variant, the analysis works for sampling without replacement when the neighboring dataset is obtained by changing a user's data, i.e., substitution (see Ullman (2017); Balle et al. (2018)).*

# B    PROMPT CONSTRUCTION FUNCTIONS FOR EACH TASK IN SECTION 5

In Tab. 5 and 6 we provide our prompt construction function $PB(\text{instruction}, \mathcal{D}^y, y)$ for each task in Section 5.

# C    DATASETS USED IN SECTION 5

**AGNews**    The AG News (AG's News Corpus) dataset (Zhang et al., 2015) consists of news articles belonging one of the 4 labels (World, Sports, Business, Sci/Tech). The AG News has 30,000 training and 1,900 test samples per class.

**TREC**    The Text REtrieval Conference (TREC) question classification dataset (Voorhees & Tice, 2000) consists of questions that have answers belonging one of the 6 answer types (Number, Location, Person, Description, Entity, Abbreviation). The TREC has in total 5,500 training and 500 test samples that are non-uniform across the labels.

Table 5: Prompt construction function $PB(\text{instruction}, \mathcal{D}^y, y)$ for each text classification task in Section 5. The prompt starts with the instruction in the first sentence and then follows a demonstration from $\mathcal{D}^y$ and finally a generation is expected from $LM$ with label $y$.

| Task | Prompt construction function $PB(\text{instruction}, \mathcal{D}^y, y)$ | Labels |
|---|---|---|
| AGNews | Given a label of news type, generate the chosen type of news accordingly.

News Type: World
Text: Australia boosts anti-terror measures at small airports SYDNEY : The Australian government announced a major security upgrade for nearly ...

News Type: World
Text: | World, Sports, Business, Technology |
| DBPedia | Given a label of document type, generate the chosen type of document accordingly.

Document Type: Company
Text: Cherry Lane Music was founded in 1960 by Milton Okun in the apartment above the Cherry Lane Theater in Greenwich Village of New York City...

Document Type: Company
Text: | Company, School, Artist, Athlete, Politician, Transportation, Building, Nature, Village, Animal, Plant, Album, Film, Book |
| TREC | Given a label of answer type, generate a question based on the given answer type accordingly.

Answer Type: Number
Text: How many people in the world speak French ?

Answer Type: Number
Text: | Number, Location, Person, Description, Entity, Abbreviation |

Table 6: Prompt construction function $PB(\text{instruction}, \mathcal{D}^y, y)$ for each information extraction task in Section 5. The prompt starts with the instruction in the first sentence and then follows a demonstration from $\mathcal{D}^y$ and finally a generation is expected from $LM$ with label $y$.

| Task | Prompt construction function $PB(\text{instruction}, \mathcal{D}^y, y)$ |
|---|---|
| MIT-G | Given a genre for the film, generate a description accordingly and make sure to include the given genre in the description.

Genre: holiday
Sentence: what is the name of this perennial holiday favorite featuring an elderly miser learning the error of his ways thanks to three ghostly visitations

Genre: action
Sentence: |
| MIT-D | Given a director for the film, generate a description accordingly and make sure to include the given director in the description.

Director: pixar
Sentence: what pixar animated film features a talking dog named dug

Director: disney
Sentence: |

**DBpedia** The DBpedia ontology classification dataset (Zhang et al., 2015) consists of contents belonging one of the 14 topics (Company, School, Artist, Athlete, Politician, Transportation, Building, Nature, Village, Animal, Plant, Album, Film, Book). The DBpedia has 40,000 training samples and 5,000 testing sample per class.

**MIT Movies** MIT Movies trivia10k13 dataset (Liu et al., 2012) consists of movie reviews that have slots (movie genre (MIT-G) and director name (MIT-D)) to be used for information extraction task. MIT-G has 2,953 training samples and 100 test samples whereas MIT-D has 1,561 training samples and 100 test samples.

## D    PROMPT FORMAT DURING ICL FOLLOWING PRIOR WORK ZHAO ET AL. (2021)

While we use the same prompt format during ICL following prior work by Zhao et al. (2021), we present them here in Tab. 7 and 8 for the convenience of the reader.

Table 7: The prompts used during ICL for text classification tasks, taken from Tab. 5 of Zhao et al. (2021).

| Task | Prompt | Labels |
|---|---|---|
| AGNews | Classify the news articles into the categories of World, Sports, Business, and Technology.

Article:    USATODAY.com- Retail sales bounced back a bit in July, and new claims for jobless benefits fell last week, the government said Thursday, indicating the economy is improving from a midsummer slump.
Answer: Business

Article:  New hard-drive based devices feature color screens, support for WMP 10.
Answer: | World, Sports, Business, Technology |
| DBPedia | Classify the documents based on whether they are about a Company, School, Artist, Athlete, Politician, Transportation, Building, Nature, Village, Animal, Plant, Album, Film, or Book.

Article:  Geoffrey D. Falksen (born July 31 1982) is an American steampunk writer.
Answer: Artist

Article:    The Perrin River is a 1.3-mile-long (2.1 km) tidal river in the U.S. state of Virginia. It is a small inlet on the north shore of the York River near that river's mouth at Chesapeake Bay.
Answer: | Company, School, Artist, Athlete, Politician, Transportation, Building, Nature, Village, Animal, Plant, Album, Film, Book |
| TREC | Classify the questions based on whether their answer type is a Number, Location, Person, Description, Entity, or Abbreviation.

Question:  How did serfdom develop in and then leave Russia?
Answer Type: Description

Question: When was Ozzy Osbourne born?
Answer Type: | Number, Location, Person, Description, Entity, Abbreviation |

Table 8: The prompts used during ICL in information extraction tasks, taken from Tab. 6 of Zhao et al. (2021).

| Task | Prompt |
|---|---|
| MIT-G | Sentence: last to a famous series of animated movies about a big green ogre and his donkey and cat friends
Genre: animated

Sentence: what is a great comedy featuring the talents of steve carell as a loser looking for a friend
Genre: |
| MIT-D | Sentence: in 2005 director christopher nolan rebooted a legendary dc comics superhero with a darker grittier edge in which movie
Director: christopher nolan

Sentence: what 1967 mike nichols film features dustin hoffman in romantic interludes with anne bancroft as mrs robinson
Director: |

# E  HYPERPARAMETER SEARCH

Hyperparameters for the main results presented in Tab. 1 are provided in Tab. 9. Hyperparameter search results for all the datasets are provided in Tab. 10-14.

Table 9: Hyperparameters for the main results presented in Tab. 1. When RVP is True, we simply take top-100 token ($K = 100$ in Alg.1) using the OpenAI API.

| Dataset | $\min_y \mathcal{D}^y_{\text{priv}}$ | M | N | RVP | $T_{\max}$ | DP | $\sigma$ for $\epsilon = 1, 2, 4, 8$ |
|---|---|---|---|---|---|---|---|
| AGNEWS | 30,000 | 10 | 2 | True | 100 | Gaussian | [0.51, 0.46, 0.39, 0.31] |
| DBPedia | 40,000 | 40 | 2 | True | 100 | Gaussian | [0.63, 0.54, 0.45, 0.36] |
| TREC | 835 | 80 | 1 | False | 15 | Gaussian | [1.36, 0.95, 0.69, 0.51] |
| MIT-G | 2,953 | 20 | 4 | True | 20 | Gaussian | [1.08, 0.81, 0.64, 0.50] |
| MIT-D | 1,561 | 20 | 4 | True | 20 | Gaussian | [1.52, 1.04, 0.77, 0.58] |

Table 10: Hyperparameter search results for the AGNews dataset for $\epsilon = 4$ on GPT-3 Babbage model with 4-shot demonstrations for ICL.

(a) RVP=True

| | $N = 1$ | $N = 2$ | $N = 4$ |
|---|---|---|---|
| $MN = 20$ | $65.0_{5.4}$ | $71.3_{4.6}$ | $65.2_{3.3}$ |
| $MN = 40$ | $68.2_{4.3}$ | $69.2_{3.6}$ | $67.4_{4.8}$ |
| $MN = 80$ | $67.4_{2.8}$ | $68.3_{1.0}$ | $67.3_{3.3}$ |

(b) RVP=False

| | $N = 1$ | $N = 2$ | $N = 4$ |
|---|---|---|---|
| $MN = 20$ | $61.7_{5.0}$ | $52.6_{6.9}$ | $49.0_{7.0}$ |
| $MN = 40$ | $64.6_{5.9}$ | $58.7_{8.1}$ | $58.5_{2.0}$ |
| $MN = 80$ | $66.1_{2.0}$ | $63.0_{5.9}$ | $53.4_{10.6}$ |

Table 11: Hyperparameter search results for the DBPedia dataset for $\epsilon = 4$ on GPT-3 Babbage model with 4-shot demonstrations for ICL.

(a) RVP=True

| | $N = 1$ | $N = 2$ | $N = 4$ |
|---|---|---|---|
| $MN = 20$ | $81.1_{3.2}$ | $81.5_{1.8}$ | $80.7_{6.8}$ |
| $MN = 40$ | $79.6_{3.8}$ | $81.5_{3.4}$ | $81.2_{5.3}$ |
| $MN = 80$ | $78.9_{1.9}$ | $83.1_{4.3}$ | $81.4_{3.6}$ |

(b) RVP=False

| | $N = 1$ | $N = 2$ | $N = 4$ |
|---|---|---|---|
| $MN = 20$ | $68.7_{2.4}$ | $64.4_{2.7}$ | $49.8_{8.4}$ |
| $MN = 40$ | $77.6_{3.4}$ | $67.9_{9.8}$ | $67.0_{9.8}$ |
| $MN = 80$ | $78.7_{3.2}$ | $81.9_{4.3}$ | $67.4_{17.8}$ |

Table 12: Hyperparameter search results for the TREC dataset for $\epsilon = 4$ on GPT-3 Babbage model with 4-shot demonstrations for ICL.

(a) RVP=True

|          | $N = 1$     | $N = 2$     | $N = 4$     |
| -------- | ----------- | ----------- | ----------- |
| $MN = 20$ | $46.2_{3.3}$ | $41.7_{3.9}$ | $42.7_{5.7}$ |
| $MN = 40$ | $46.9_{4.2}$ | $46.4_{3.8}$ | $45.6_{6.3}$ |
| $MN = 80$ | $49.2_{6.0}$ | $47.2_{6.0}$ | $44.7_{2.6}$ |

(b) RVP=False

|          | $N = 1$     | $N = 2$     | $N = 4$     |
| -------- | ----------- | ----------- | ----------- |
| $MN = 20$ | $46.8_{7.9}$ | $44.8_{2.7}$ | $43.6_{9.4}$ |
| $MN = 40$ | $48.8_{2.2}$ | $44.0_{3.5}$ | $43.8_{4.7}$ |
| $MN = 80$ | $50.4_{5.5}$ | $47.0_{5.0}$ | $45.4_{4.0}$ |

Table 13: Hyperparameter search results for the MIT-G dataset for $\epsilon = 4$ on GPT-3 Babbage model with 4-shot demonstrations for ICL.

(a) RVP=True

|          | $N = 1$     | $N = 2$     | $N = 4$     |
| -------- | ----------- | ----------- | ----------- |
| $MN = 20$ | $53.0_{2.3}$ | $47.5_{5.1}$ | $35.1_{5.6}$ |
| $MN = 40$ | $53.4_{5.4}$ | $54.1_{4.6}$ | $45.7_{7.3}$ |
| $MN = 80$ | $53.1_{5.3}$ | $51.1_{5.2}$ | $54.7_{5.4}$ |

(b) RVP=False

|          | $N = 1$     | $N = 2$      | $N = 4$      |
| -------- | ----------- | ------------ | ------------ |
| $MN = 20$ | $46.5_{7.0}$ | $23.0_{10.5}$ | $12.0_{2.9}$  |
| $MN = 40$ | $51.2_{5.2}$ | $55.4_{3.5}$  | $31.9_{14.2}$ |
| $MN = 80$ | $53.2_{4.2}$ | $51.8_{5.9}$  | $50.5_{9.8}$  |

Table 14: Hyperparameter search results for the MIT-D dataset for $\epsilon = 4$ on GPT-3 Babbage model with 4-shot demonstrations for ICL.

(a) RVP=True

|          | $N = 1$     | $N = 2$     | $N = 4$      |
| -------- | ----------- | ----------- | ------------ |
| $MN = 20$ | $71.5_{6.6}$ | $70.8_{7.7}$ | $65.1_{10.1}$ |
| $MN = 40$ | $70.2_{5.9}$ | $71.4_{7.6}$ | $68.2_{8.5}$  |
| $MN = 80$ | $69.8_{5.9}$ | $70.5_{6.0}$ | $74.6_{4.2}$  |

(b) RVP=False

|          | $N = 1$     | $N = 2$     | $N = 4$       |
| -------- | ----------- | ----------- | ------------- |
| $MN = 20$ | $69.7_{4.1}$ | $9.2_{8.5}$  | $5.7_{2.7}$    |
| $MN = 40$ | $72.2_{3.2}$ | $66.8_{6.3}$ | $9.9_{5.2}$    |
| $MN = 80$ | $71.4_{3.2}$ | $78.0_{2.2}$ | $40.7_{19.16}$ |

## F  EMPIRICAL PRIVACY EVALUATION BY MEMBERSHIP INFERENCE ATTACKS

While differential privacy provides the theoretical guarantee to privacy leakage, it is also important to evaluate the empirical privacy leakage (Blanco-Justicia et al., 2022). To better understand the privacy protection of our DP few-shot generation for in-context learning, we empirically evaluate the privacy of our methods by membership inference attack (MIA) (Shokri et al., 2017), that is widely used for estimating the practical privacy leakage.

We follow the prior work (Duan et al., 2023) that instantiates MIA in the framework of in-context learning. The goal of their attack is to determine if a given data point was used within the prompt of the LLM. Perhaps expectedly, using actual samples from the private dataset leads to successful MIA results. To apply the MIA for our DP few-shot generation, we take the following approach. We study the DBPedia dataset and split the dataset in two parts for member and non-member samples. We use the member samples to generate synthetic demonstrations with our DP algorithm. Similar to Duan et al. (2023), we consider 1-shot ICL on the Babbage model and generate 1-shot demonstrations in 5 independent runs for $\epsilon = [1, 2, 4, 8]$. For MIA against DP few-shot generation, we make queries for 50 samples from members and 50 from non-members, and calculate the TPR/FPR. For each DP generated demonstrations, we run 20 trials for MIA and we finally average across the 100 trials to calculate the averaged AUC for each $\epsilon$. For the MIA against non-private 1-shot ICL, we follow Duan et al. (2023), the 1-shot in prompt is the member sample and we sample 50 non-member samples. We repeat 100 trials and compute the averaged AUC.

We present the membership privacy attack result in Tab. 15. Similar to Duan et al. (2023), we observe that using actual samples from the private dataset leads to successful MIA results (81.84 corresponding to $\epsilon = \infty$). On the other hand, our DP solution reduces the AUC of MIA to almost

Table 15: Empirical privacy evaluation for 1-shot ICL by MIA on Babbage model.

| $\epsilon$ | 1 | 2 | 4 | 8 | $\infty$ |
|---|---|---|---|---|---|
| AUC | 50.56 | 50.58 | 50.55 | 50.53 | 81.84 |

random guesses, demonstrating the effectiveness of our privacy-preserving methodology for ICL. It is interesting to observe that even for a theoretically large epsilon value such as 8, MIA attack remains close to random guess, which presents promising (empirical) privacy-utility tradeoffs.

## G  MONETARY COST

We provide the monetary cost of running our Alg. 1. For each token generation, we make $M$ API calls in Alg. 1. For each sample, we generate at most $T_{max}$ tokens. The total number of token processed by the API calls is proportional to $M \times \sum_{t=P}^{P+T_{max}} t$ tokens, where $P$ is the (expected) prompt length. We could estimate $P$ by computing the average of token length in the training set and the token length of additional instruction as shown in Tab. 5 and Tab. 6.

We provide the estimation of monetary cost for one-shot generation for DBPedia on different models (Ada, Babbage, Curie, Davinci) in Tab. 16.

Table 16: Estimated cost for 1-shot generation in Alg. 1 for DBPedia on different models.

| Model | Ada | Babbage | Curie | Davinci |
|---|---|---|---|---|
| Cost ($) | 0.36 | 0.45 | 1.81 | 18.09 |

We point out that this process represents a one-time cost that is incurred prior to the deployment of the model.

## H  DEMONSTRATIONS

We provide the DP few-shot generations at $\epsilon = 4$ for DBPedia by the Babbage model in Tab. 17. We provide a brief observation for the quality of the demonstrations.

While the Babbage model is comparatively smaller and less advanced than the current state-of-the-art models, its synthetic generations maintain a reasonable level of coherence and fluency. The length of these synthetic generations generally aligns with the samples in their respective training datasets. However, we have noticed that longer generations occasionally result in the repetition of the last sentence. We also noticed factual errors when inspecting the synthetic generations in detail. We believe that these issues can be mitigated by employing more capable instruction-following models, refining the instruction during the generation process and advances in hallucination mitigation.

Table 17: Examples of DP few-shot generations on DBPedia.

| DP few-shot generations | Label |
|---|---|
| The company was founded in 1883 by a group of German immigrants in the United States. The company was originally known as the American Electric Company. | Company |
| The School of the Arts is a school of the University of the Arts in Philadelphia, Pennsylvania. It is a graduate school of the University of the Arts. It is located on the main campus of the University of the Arts. The School of the Arts is a member of the Association of Independent Schools of the United States and Canada. | School |
| The Beatles were an English rock band formed in Liverpool in 1960. The group consisted of John Lennon, Paul McCartney, George Harrison, and Ringo Starr. The Beatles are widely considered to be one of the most successful and influential bands in the history of popular music. They are also one of the most commercially successful bands of all time, with over 600 million records sold worldwide. The Beatles have been inducted into the Rock and Roll Hall of Fame, the Songwriters Hall of Fame, and the | Artisit |
| The Athlete is a person who is physically fit and able to perform at a high level of performance. The term is used in sports to describe a person who is physically fit and able to perform at a high level of performance. | Athlete |
| The political party of the President of the United States is the Democratic Party. The Democratic Party is the oldest political party in the United States. The Democratic Party was founded in 1828 by Thomas Jefferson and James Madison. | Politician |
| The train was a long-distance passenger train that operated in the United States from 1869 to 1891. It was the first passenger train to operate in the United States. The train was built by the New York Central Railroad and was named the "Empire Builder". The train was built to carry passengers from New York to Chicago, and was the first train to cross the continent. | Transportation |
| The building was built in 1892 by the architect John H. C. Smith. It was designed in the Italianate style and is a two-story, five bay, rectangular building with a hipped roof. The main entrance is on the second floor. The building is constructed of brick and stone. The building is a contributing property to the Old Town Historic District. The building was listed on the National Register of Historic Places in 1982. The building is located at the corner of Main and Main | Building |
| The world's largest and most diverse collection of tropical rainforests is found in the Amazon Rainforest. The Amazon Rainforest is the largest tropical rainforest in the world. It is located in the western Amazon Basin in South America. | Nature |
| The village of Kishinev is located in the Kishinev Oblast of the Russian Federation. It is located on the Kishinev River, a tributary of the Dnieper River. | Village |
| The animal is a small, brown, furry, and very active mammal. It is found in the forests of the northern hemisphere. It is a member of the family Cebidae. It is a very common species. | Animal |
| The plant is a small, herbaceous perennial plant with a long, narrow, erect stem. The leaves are alternate, simple, and entire. The flowers are produced in a raceme, and are produced in the axils of the leaves. The flowers are white, pink, or purple, and are produced in the axils of the leaves. The fruit is a small, dry, woody capsule. The plant is native to the Mediterranean region. It is found in the Mediterranean region, | Plant |
| The album was released on November 1, 2009. It was recorded in the studio in Los Angeles, California, and was produced by the band's guitarist, John Hanes. | Album |
| The film is a documentary about the life of the late singer and songwriter, John Lennon. It was directed by Martin Scorsese and produced by Albert Maysles. The film was shot in black and white and was shot in the United States, England, and Germany. It was released in the United States on November 30, 1980. The film was nominated for the Academy Award for Best Documentary Feature. The film was also nominated for the Academy Award for Best Documentary Short Subject. | Film |
| The book is a novel by the American author John Grisham. It is the first novel in the legal thriller series The Firm. The book was published in the United States on September 1, 1992 by Doubleday, an imprint of Random House. The book was first published in the United Kingdom by Jonathan Cape on September 1, 1992. The book was later published in the United States by Random House in September 1993. | Book |

