# OpenReview forum: "Privacy-Preserving In-Context Learning with Differentially Private Few-Shot Generation"
_ICLR.cc/2024/Conference — ICLR 2024 poster_

### Official Review · Reviewer_95y5 · 2023-10-31

**Soundness:** 3 good
**Presentation:** 3 good
**Contribution:** 3 good
**Rating:** 8
**Confidence:** 2

**Summary:**

This paper studies an emergent and important problem of how to generate private demonstrations for in-context learning. The authors proposed to generate synthetic few shot examples from the private database (Algorithm 1). As a result of post-processing property of DP, number of queries does introduce extra privacy cost.

**Strengths:**

Studies an important problem and provide intuitive solution. Theoretical analysis looks correct to me.

The experiments compared with a strong competitor (asking LLM to generate its own demonstrations). Moreover. this finding is also interesting in its own sense (with 20-30% accuracy improvement in many cases.)! Private ICL shows comparable accuracy to the non-private  few-shot prompt accuracy.

**Weaknesses:**

Algorithm 1 seems to incur M foundation model api calls. How much monetary cost does the algorithm incur?

In the paper, the author investigated report noisy max with exponential mechanism. Will using RNM with gaussian mechanism lead to better accuracy?

**Questions:**

See weakness.

---

> ### Author Response · Authors · 2023-11-18
> **Response by Authors**
>
> We thank the reviewer for their careful reading and thoughtful comments along with their positive score. We address all their questions below.
>
> 1. As the reviewer correctly pointed out, indeed Algorithm 1 incurs M foundation model API calls per token generation. Along with M, the total cost of a synthetic sample generation also depends on the number of tokens in the prompts and the number of generated tokens. We present the monetary costs associated with applying Algorithm 1 for the results in Table 3 to provide readers with a clear understanding of the expenses incurred for different models, including Ada, Babbage, Curie, and Davinci.  We highlight that these monetary costs represent one-time expenses incurred prior to the deployment of the model. We add the discussion on monetary cost in Appendix G and  would be happy to engage in further discussions with the reviewer should any follow-up suggestions arise.
>
>
>
>
> | Model | Monetary cost of Algorithm 1 |
> |----------|----------|
> | Ada    | $ 0.36  |
> | Babbage    | $ 0.45  |
> | Curie    | $ 1.81  |
> | Davinci    | $ 18.09  |
>
>
> 2. In our paper, we investigate two mechanisms: (i) Gaussian DP mechanism and (ii)  Report-Noisy-Max with Exponential DP mechanism. We compare both mechanisms in Table 4 and observe in general competitive performances between the two DP mechanisms. For both mechanisms we use the `argmax` sampling (i.e., report-noisy-max) to generate tokens. Therefore, we can state that they perform similarly in our experiments.
>
> However there is an important difference between the two mechanisms. Report-Noisy-Max with Exponential DP mechanism can only work with `argmax` sampling to generate tokens, however, line 14 of Algorithm 1 can be utilized to improve the maximum probability to be 1 thanks to the fact that only the `argmax` token is reported. Unfortunately, this benefit cannot be instantiated with the Gaussian DP mechanism. In other words,  Report-Noisy-Max with Exponential DP works as long as sensitivity is bounded in \ell_\infinity norm. But Gaussian Mechanism (even if only reporting `argmax` token, i.e, RNM with Gaussian Noise) requires the sensitivity to be bounded in \ell_2 norm, which is a stricter requirement. However, the advantage of the Gaussian DP mechanism is that it benefits from the post-processing of DP, therefore, more advanced sampling techniques for token generation can be used instead of the basic `argmax` sampling. We leave the exploration of this to future work.

---

### Official Review · Reviewer_MYhF · 2023-10-31

**Soundness:** 3 good
**Presentation:** 3 good
**Contribution:** 3 good
**Rating:** 8
**Confidence:** 3

**Summary:**

To prevent the leakage of private data, the current approach necessitates consulting the large model only in a zero-shot manner. In this article, the author proposes a solution by generating multiple synthetic prompts based on private data. This enables leveraging the performance boost afforded by few-shot learning while maintaining privacy guarantees.

**Strengths:**

In the current industry practice, safeguarding privacy data requires organizations to pre-train or fine-tune large models, demanding substantial resources. Leveraging synthetic data as prompts to enhance large model capabilities not only preserves privacy but also significantly reduces resource consumption.  Adding noise to the probability of generating the next word in a large model is a relatively novel approach.

**Weaknesses:**

Generating a composite prompt requires consulting the large model M*L times, leading to a significant consumption of resources.
  + The necessity of employing an untrusted model to generate the next word poses a challenge. While open-source large models are an option, their performance in generation tasks tends to be lower compared to closed-source counterparts. There is some uncertainty surrounding the final quality of the synthesized prompt, including aspects such as length, coherence, and overall fluency.

**Questions:**

Pls specify the quality of the synthesized prompt.

---

> ### Author Response · Authors · 2023-11-18
> **Response by Authors**
>
> We thank the reviewer for their careful reading and thoughtful comments along with their positive score.
>
> In response to the reviewer's comment about the resource consumption involved in generating a composite prompt by consulting the large model M*L times, we acknowledge and agree with this observation. However, it is crucial to highlight that this process represents a one-time cost that is incurred prior to the deployment of the model. Once this initial stage is completed, it enables the model to be used indefinitely without incurring additional costs in this regard.
>
> We recognize the concern raised about the reliance on an untrusted model to generate synthetic few-shot demonstrations. It may be the case that open-source models might not always match the generation capabilities of their closed-source counterparts. Nonetheless, it's important to highlight the rapid advancements being made in the capabilities of open-source models. Recent progress (with open-source LLMs such as LLama-2, Falcon, Vicuna, Mistral, among others) indicates substantial improvements in these models, suggesting a promising future where they could potentially rival the performance of their closed-source counterparts. We also want to emphasize that investigating the use of open-sourced models represents a crucial and intriguing avenue for future research. Delving into this area could effectively address the issue of reliance on an untrusted model.
>
> We thank the reviewer for their question regarding the quality of the synthesized prompt. In this work, our primary focus is on in-context learning, and we believe that the performance of models with synthetic few-shot demonstrations during this process is a strong indicator of the prompt's quality. However, we concur with the reviewer that aspects such as length, coherence, and fluency are critical in determining the overall quality of the synthesized prompt. While a comprehensive evaluation of these aspects slightly exceeds the scope of our current study, which is centered on in-context learning, we acknowledge the importance of these factors. In response to the reviewer's question, we additionally include sample few-shot generations by Babbage from randomly selected experiments in Appendix H of revised manuscript. These samples provide additional insights into the quality of the synthesized prompts, particularly in relation to the aspects highlighted by the reviewer. We note that while the Babbage model is comparatively smaller and less advanced than the current state-of-the-art models, its synthetic generations maintain a reasonable level of coherence and fluency. Generally, the length of these synthetic generations aligns with the samples in their respective training datasets. However, we have noticed that longer generations occasionally result in the repetition of the last sentence. We believe that this issue can be mitigated by employing more capable instruction-following models and refining the instruction during the generation process.

---

### Official Review · Reviewer_vhzg · 2023-10-31

**Soundness:** 3 good
**Presentation:** 3 good
**Contribution:** 4 excellent
**Rating:** 8
**Confidence:** 3

**Summary:**

This paper studies how to incorporate differential privacy into in-context learning with large language models. The authors propose a DP algorithm which generates synthetic few-shot examples from a private dataset; these “demonstrations” can then be used downstream for in-context learning without incurring additional privacy costs. The authors evaluate their algorithm across several benchmark datasets and privacy regimes, including a fully private ($\epsilon=0$) case which doesn’t use the private dataset at all.

**Strengths:**

1. The empirical evaluation (including an ablation study) is very thorough.
2. Compared to previous / concurrent work, I think the “generating synthetic few-shot data” approach is really practical since (due to post-processing) answering any number of queries now doesn’t affect the privacy guarantee.
3. The proposed framework is flexible enough to adapt to many potential use cases, and I could see that there could be interesting future work down the line.

**Weaknesses:**

None of the experimental baselines compare the proposed algorithm to existing work. In particular, I think it could be instructive to see how DP few-shot generation compares to DP fine-tuning approaches.

**Questions:**

It seems likely the the PATE-like component of Algorithm 1 could (similarly to PATE) have a stronger data-dependent privacy analysis if the models “agree.” While by no means necessary, I would be interested to see this!

---

> ### Author Response · Authors · 2023-11-18
> **Response by Authors**
>
> We thank the reviewer for their careful reading and thoughtful comments along with their positive score. We offer the following detailed discussion in response to the reviewer's comments.
>
> We agree with the reviewer that researching the comparative effectiveness of DP few-shot generation versus DP fine-tuning is a compelling and important area of study. The appropriateness of either approach may hinge on a variety of factors, including the nature of the downstream task and the level of access to the model. For instance, direct fine-tuning with DP might not be feasible in scenarios where one is limited to API-level interactions with the model. Conversely, certain downstream tasks might not yield optimal results with mere few-shot demonstrations, potentially benefiting more from fine-tuning for enhanced adaptation to specific domains. In light of these considerations, we recognize the importance of future research aimed at elucidating the circumstances under which DP few-shot generation or DP fine-tuning would be more advantageous.
>
> We thank the reviewer for the insightful comment regarding the potential of a stronger data-dependent privacy analysis similar to the prior work on PATE that utilizes the level of agreement among the teachers. Indeed this improvement would be beneficial for Algorithm 1 because to obtain strong privacy levels (i.e. low epsilon values), we apply resampling the demonstrations from the private dataset for each token generation that benefits the amplification of privacy by subsampling theorem. Therefore, stronger data-dependent privacy analysis may benefit overcoming the need for resampling per token generation and in general provide us with improved privacy levels. We agree that this presents an intriguing direction for future research.

---

### Official Review · Reviewer_kLtt · 2023-11-04

**Soundness:** 3 good
**Presentation:** 3 good
**Contribution:** 3 good
**Rating:** 8
**Confidence:** 4

**Summary:**

This study addresses the challenges of in-context learning (ICL) with large language models (LLMs) on private datasets, focusing on privacy concerns. The researchers propose an algorithm that generates synthetic few-shot demonstrations with formal differential privacy (DP) guarantees, enabling effective ICL while protecting private information in prompts. Their empirical experiments demonstrate that this approach maintains competitive performance with strong privacy levels. Additionally, they explore zero-shot solutions where LLMs generate their own demonstrations, showing potential for achieving privacy without compromising performance.

**Strengths:**

1)The paper studies simple approach of achieving privacy in ICL (synthetic data generation and use it as demonstrations)

2)Paper for most parts is clearly written and is easy to read.

**Weaknesses:**

1) No comparison with real-world threat models has been provided. Epsilon-utility trade-offs can be misleading without testing them against actual attacks, as epsilon guarantees are built upon numerous assumptions, as indicated in [1, 2, 3]. For a comprehensive evaluation, it is essential to conduct experiments that demonstrate trade-offs between  empirical privacy and utility.


2) *"*..A different line of work (Feyisetan et al., 2020; Xu et al., 2020; Du et al., 2023) focuses on sanitizing user texts locally before releasing them to the server based on metric local differential privacy (Chatzikokolakis et al., 2013). Such methods usually incur huge overheads to the utility of the sanitized text ..*"* This statement is not necessarily true and it should be revised to avoid making broad generalizations. As evidenced in a recent study [3], the use of a language model prompted "zero-shot" to generate paraphrases exhibited a clear and significantly better (empirical)privacy-utility tradeoffs.


**Refs**

[1] A Critical Review on the Use (and Misuse) of Differential Privacy in Machine Learning (https://arxiv.org/abs/2206.04621)

[2] TEM: High utility metric differential privacy on text. (https://arxiv.org/abs/2107.07928)

[3] Locally Differentially Private Document Generation Using Zero Shot Prompting (https://arxiv.org/abs/2310.16111 )

**Questions:**

See above

---

> ### Author Response · Authors · 2023-11-18
> **Response by Authors**
>
> We thank the reviewer for their careful reading and thoughtful comments. We address all their questions below.
>
> 1. Thank you for bringing to our attention the important aspect of testing our results against real-world threat models. We acknowledge the reviewer's perspective regarding empirical evaluations that test epsilon-utility trade-offs against actual attacks as such assessments lead to a more holistic understanding of privacy and utility.
>
> To address this, we follow the prior work (Duan et al. 2023) where the authors instantiate membership inference attack (MIA) in the framework of in-context learning (ICL). The goal of their attack is to determine if a given data point was used within the prompt of the LLM. Perhaps expectedly, using actual samples from the private dataset leads to successful MIA results. To apply MIA for our DP solution, we take the following approach. We study the DBPedia dataset and split the dataset in two parts for member and non-member samples. We use the member samples to generate synthetic demonstrations with our DP algorithm. Similar to Duan et al, we consider 1-shot ICL on the Babbage model and generate 1-shot demonstrations in 5 independent runs for $\epsilon=[1, 2 ,4 ,8]$. During ICL, we attempt MIA for the samples from members and non-members and similarly calculate the TPR/FPR. For each DP generated demonstrations, we run 20 trials for MIA and we finally average across the 100 trials to calculate the AUC for each epsilon. For the MIA against non-private 1-shot ICL, we follow Duan et al.(2023) and average across 100 trials for AUC.
>
> We present the membership privacy attack result with the AUC metric in the table below.
>
> | $\epsilon$ | 1 | 2 | 4 | 8 | $\infty$ |
> |---|---|---|---|---|---|
> | MIA AUC | 50.56 | 50.58 | 50.55 | 50.53 | 81.84 |
>
> Similar to Duan et al. (2023), we observe that using actual samples from the private dataset leads to successful MIA results (81.84 corresponding to $\epsilon=\infty$). On the other hand, our DP solution reduces the MIA AUC to almost random guesses, demonstrating the effectiveness of our privacy-preserving methodology for ICL. It is interesting to observe that even for a theoretically large epsilon value such as 8, MIA attack remains close to random guess, which presents promising (empirical) privacy-utility tradeoffs. We add the detailed empirical privacy evaluation by MIA in Appendix F in the revised draft.
>
> 2. We thank the reviewer for pointing out this statement in our paper. We acknowledge the importance of accurate discussion and appreciate the reference to the recent study the reviewer provided. We propose the following modification (along with adding more related work) and we would be happy to revise this in further discussions with the reviewer should any follow-up suggestions arise:
>
> “We finally review an active area of research on the generation of sanitized text documents. In this domain, Feyisetan et al. (2020); Xu et al. (2020); Carvalho et al. (2021); Du et al. (2023) add noise at the word level and use metric local differential privacy (Chatzikokolakis et al., 2013). Mattern et al. (2022b); Utpala et al. (2023) directly operate on documents using local differential privacy (LDP) (Kasiviswanathan et al., 2010; Duchi et al., 2013) where the former is based on fine-tuning for paraphrasing and the latter uses zero-shot prompting for paraphrasing followed by sanitization.”
>
> We would very much appreciate the reviewer considering increasing their rating in case they find our responses compelling.

---

> > ### Comment · Reviewer_kLtt · 2023-11-18
> > **Thanks for the reply**
> >
> > Thanks for additional experiments with MIA and modifications about DP mechanism for sanitization. I have now increased my score.

---

> > > ### Author Response · Authors · 2023-11-20
> > > **Response by Authors**
> > >
> > > We thank the reviewer for their participation in the discussion. We appreciate your time, comments, and valuable suggestions.

---

### Meta-Review · Area_Chair_v8NG · 2023-12-05

**Metareview:**

The paper studies the problem of in-context learning on private datasets, with the constraint of differential privacy. The paper proposes a novel algorithm that generates synthetic few-shot demonstrations with differential privacy, using either the Gaussian mechanism or Report Noisy-Max mechanism. It provides an empirical evaluation of the algorithm showing that it has strong performance even in the high privacy regime. Overall, the paper proposes a nice algorithm for a timely problem and is a good fit for ICLR.

**Justification For Why Not Higher Score:**

The algorithm is relatively straightforward and the novelty not significant enough to justify a higher score.

**Justification For Why Not Lower Score:**

The problem is timely, the proposed algorithm is nice, and the paper opens up some interesting new directions for future work to investigate.

---

### Decision · Program_Chairs · 2024-01-16

Accept (poster)